# Reliability and Validity of Scoliosis Measurements Obtained with Surface Topography Techniques: A Systematic Review

**DOI:** 10.3390/jcm11236998

**Published:** 2022-11-26

**Authors:** Xinyu Su, Rui Dong, Zhaoyong Wen, Ye Liu

**Affiliations:** School of Sport Science, Beijing Sport University, Beijing 100084, China

**Keywords:** scoliosis, assessment, surface topography, reliability, validity, reproducibility, posture

## Abstract

Background. Surface topography (ST) is one of the methods in scoliosis assessment. This study aimed to systematically review the reliability and validity of the ST measurements for assessing scoliosis. Methods. A literature search of four databases was performed and is reported following PRISMA guidelines. The methodological quality was evaluated using Brink and Louw appraisal tool and data extraction was performed. The results were analyzed and synthesized qualitatively using the level of evidence method. Results. Eighteen studies were included and analyzed. Four were evaluated for reliability, six for validity, and eight for reliability and validity. The methodological quality of fourteen studies was high. Good to excellent intra-investigator reliability was shown on asymmetry, sagittal, horizontal, and most frontal ST measurements (evidence level: strong). Asymmetry and most frontal, sagittal, horizontal ST measurements showed good to excellent inter-investigator reliability (evidence level: moderate). When comparing corresponding ST and radiological measurements, good to strong validity was shown on most frontal, sagittal, and asymmetry measurements (evidence level: strong). Formetric measurements had good intra-investigator reliability and validity (evidence level: strong). Conclusions. Most asymmetry, sagittal, and frontal ST measurements showed satisfactory reliability and validity. Horizontal ST measurements showed good reliability and poor validity. The ST technique may have great potential in assessing scoliosis, especially in reducing radiation exposure and performing cosmetic assessments.

## 1. Introduction

Scoliosis is a three-dimensional structural deformity of the spine, including abnormal alignment of the vertebrae in the coronal, sagittal, and horizontal planes. According to the different causes, scoliosis can be categorized into several main types [1]: Idiopathic scoliosis (IS), congenital scoliosis (CS), neuromuscular scoliosis (NS), and scoliosis from miscellaneous causes. The most common type is idiopathic scoliosis (IS), and 80% or more [2] of idiopathic scoliosis occurs in adolescents, called adolescent idiopathic scoliosis (AIS). The US Preventive Services Task Force (USPSTF) states that [3] the prevalence of adolescent idiopathic scoliosis in children and adolescents aged 10–16 years is 1% to 3%. Symptoms associated with scoliosis can be influenced by the cause of the disease, the degree of scoliosis, and its progression. Specific symptoms include abnormal appearance, back pain, and psychological problems. In severe cases, it can also lead to respiratory distress, cardiopulmonary dysfunction, and even paralysis [4,5,6,7,8].

Therefore, early detection of scoliosis and timely intervention are essential. The gold standard in diagnosing and evaluating scoliosis is the radiological method, in which the Cobb angle is measured on a spine X-ray. A Cobb angle greater than 10° is the criterion for diagnosing scoliosis. This method is highly accurate and widely used, but some disadvantages cannot be ignored. The first is that radiation exposure may lead to health problems. One study showed that patients required twenty-two X-rays during the 3 years of treatment with the Milwaukee brace [9]. In addition, repeated X-ray exposure may increase cancer risk in patients [10]. A retrospective cohort study [11] of 5573 female patients with scoliosis in the United States showed that the incidence of breast cancer in the subjects was nearly two times higher than healthy individuals. Furthermore, although the morphology of the spine can be shown on radiographs, the Cobb angle is a two-dimensional indicator that does not fully quantify the degree of three-dimensional deformity of the spine.

The drawbacks mentioned above have triggered the development of several radiation-free, three-dimensional methods, such as CT [12,13,14], ultrasound [15,16,17,18], and surface topography technique [19,20,21]. Surface topography is more widely used among these methods since it is low-cost and user-friendly. Commonly used surface topography systems include InSpeck, Quantec, Milwaukee, and Formetric 4D [22]. As a clinical measurement tool, it is crucial to determine the validity and reliability of its measurement. Some of the previous relevant studies focused on reliability and validity, but a systematic review of this literature is lacking. 

Therefore, this study aimed to systematically review the reliability and validity of the surface topography measurements for assessing scoliosis. Since various indices can be obtained through the ST technique, we classify the ST measurements into four categories: Frontal, sagittal, horizontal, and asymmetry measurements to achieve more focused results. Frontal, sagittal, and horizontal measurements are those where the anatomical landmarks used in the measurement are located in the corresponding planes. Asymmetry measurements reflect body asymmetry and usually involve more than one plane or more than one body region. Additional analyses were performed regarding the type of scanning (back-only or full-torso) and the automation level (manual or automatic identification of marker points). In terms of reliability, we analyzed intra-investigator and inter-investigator reliability to verify the reproducibility of ST measurements. Radiological measurements are considered the gold standard (especially the Cobb angle) and are well-established and widely used in all phases of scoliosis diagnosis, treatment, and management. It is of great interest to physicians and researchers how ST measurements relate to radiological measurements and in what ways and to what extent ST measurements can replace radiological measurements. Therefore, for validity evaluation, we chose radiological measurements as the reference standard and evaluated the validity by the correlation between ST measurements and radiological measurements.

## 2. Materials and Methods

### 2.1. Search Strategy

This study was based on the Preferred Reporting Items for Systematic Reviews and Meta-Analyses (PRISMA) guidelines [23]. A systematic search was conducted on four databases (Web of Science, Embase, PubMed, Cochrane Library). We combined search terms and MESH terms in a search strategy developed for PubMed, and adapted this strategy for the other databases. The search strategy of PubMed is shown in Table 1. The search language is English, and a manual search was performed on the references of the included studies.

### 2.2. Eligibility Criteria

The inclusion criteria are as follows:Subjects were patients with idiopathic scoliosis;The analysis of the patient’s scoliosis was performed using the surface topography technique;The reliability and/or validity of the surface topography measurements was evaluated;The surface topographic and radiological data were compared in the validity evaluation.

The exclusion criteria are as follows:Subjects had a history of previous spinal surgery or other spinal diseases;Subjects with scoliosis of neuromuscular, degenerative or other diagnosable cause;Subjects received spinal surgery or other treatment (including spinal orthoses and exercise therapy, etc.) during the experiment;Repeatedly published or unavailable full-text literature.

The literature search and screening were conducted by two reviewers independently following the search strategy and eligibility criteria described above. The titles and abstracts of the literature were first reviewed to exclude literature that clearly did not meet the requirements. The full text was read for literature that could not be identified for inclusion or exclusion. The study was included in the systematic review when both reviewers agreed that the study met the inclusion criteria. A third reviewer was consulted in case of any disagreement between the two reviewers during the literature search and screening process.

### 2.3. Data Extraction

The content of the data extraction included subject information, surface topography system, acquisition protocol, main evaluation indicators, and results of reliability and validity (including the measurements and methods used to obtain reference measurements used in validity analysis). Two reviewers performed data extraction independently, and the extracted contents were recorded in an abstraction form. A third reviewer was consulted in case of any disagreement.

### 2.4. Quality Assessment

Methodological quality assessment was performed using the reliability and validity critical appraisal tool proposed by Brink et al. [24]. The tool consists of thirteen items, of which five items relate to both validity and reliability studies, four items to validity studies only, and four items to reliability studies. To reduce the influence on reliability and validity results by the subject’s scoliosis progression, in our review, the appropriate interval between repeated reliability measurements and the index and standard tests was considered within 7 days. Each item could be scored as “yes”, “no” or “not applicable”. As stated in previous studies [25,26,27,28], if a study scored ≥60%, the methodological quality of the study could be considered as high; otherwise, it is considered low. Quality assessment was performed independently by two reviewers using the above-mentioned assessment tools, and any disagreements were resolved by discussion.

### 2.5. Data Analysis

The included studies showed significant heterogeneity in study design, ST measurements, and outcome measures. A meta-analysis was not performed since no three or more studies involving the same ST measurements were found. Therefore, descriptive analyses and synthesis were conducted, and heterogeneity was examined quantitatively and qualitatively using levels of evidence. The levels of evidence were divided into four categories: Strong, moderate, limited, and conflicting [29]. The specific criteria are shown in Table 2.

We classified the ST measurements into four types: Asymmetry, horizontal, sagittal, and frontal measurements (asymmetry measurements reflect body asymmetry and usually involve more than one plane or more than one body region). By categorizing similar measurements in different studies, the reliability and validity of ST measurements can be analyzed more specifically. In addition, analysis in terms of scan type (back-only or full-torso) and the automation level (manual or automatic identification of marker points) is considered.

For reliability results, we focused on intra- and inter-investigator reliability. When evaluating reliability, studies used intra-class correlation coefficient (ICC), Pearson or Spearman correlation coefficient (r), standard error of measurement (SEM), smallest detectable change (SDC), limits of agreement (LoA) or coefficient of variation (CV) to indicate that reliability results were considered adequate. It is important to note that studies using CV and correlation coefficients also need to state that there are no systematic differences (e.g., by *t*-test); otherwise, the study’s score in the quality assessment was reduced.

For validity results, the main focus is on criterion validity, which is the evaluation of validity by correlation with a reference standard (radiological parameter). Therefore, the use of Pearson or Spearman correlation coefficient (r) to express validity results is considered appropriate.

The interpretation criteria of ICC [30], r [20], and CV [31] are shown in Table 3.

## 3. Results

### 3.1. Articles Selection

According to the inclusion and exclusion criteria, eighteen articles [20,21,32,33,34,35,36,37,38,39,40,41,42,43,44,45,46,47] were finally included in this systematic review. Four [20,33,41,44] were evaluated for reliability only, six [21,35,36,40,43,45] for validity only, and eight [32,34,37,38,39,42,46,47] for both reliability and validity. The flow chart of literature screening is shown in Figure 1.

### 3.2. Quality Assessment

Of the eighteen included studies, fourteen studies were rated as high-quality (score greater than 60%). Of the eight studies in which both reliability and validity evaluations were conducted, seven [32,34,37,38,42,46,47] were of high-quality. One [41] of the four studies evaluated for reliability only was of high-quality. Of the six studies that underwent validity evaluation only, all were of high-quality. The results of the quality assessment are listed in Table 4. The weak quality assessment is mainly due to the inappropriate or lack of description of the following items: Clarified the qualifications of testers, intra-raters blinded, varied examination order, appropriate period between the reference standard and index test. 

### 3.3. Study Characteristics

After data extraction, the study characteristics of eighteen studies are listed in Table 5. 

#### 3.3.1. The Surface Topography Systems

The eighteen studies used seven different surface topography systems (we considered self-designed systems in the same category). Of these, six studies [36,39,40,42,45,46] used in self-designed systems. The Formetric system was used in six studies [32,34,38,41,43,47,48], and the BIOMOD^®^L system was used in two studies [20,33]; other systems were reported in only one study each. In terms of the measurement area, three [36,37,44] studies were used in systems (the InSpeck system, Vitus Smart Scanner, and a self-designed system) that measured the full torso and the remaining systems (e.g., Formetric, BIOMOD^®^L) measured the back only. 

#### 3.3.2. Participants

Seventeen studies [20,21,32,34,35,36,37,38,39,40,41,42,43,44,45,46,47] stated the age of the subjects. All seventeen studies involved children or/and adolescents or/and young adults younger than 25 years of age. In most studies, subjects younger than 18 years were used. Regarding the sex of the subjects, two [36,39,49] of the eighteen included studies did not specify the sex of the subjects. The remaining sixteen studies included male and female subjects and more female than male subjects, which is consistent with gender differences in the onset of scoliosis.

In terms of scoliosis angle, three [21,32,44] studies did not specify the scoliosis angle of the subjects. Of the fifteen studies that specified the scoliosis angle of the subjects, most of them had subjects with mild, moderate, and severe scoliosis, and only the subjects used in the study by Gorton G. E. [37] et al. were patients with only severe scoliosis (Cobb angle 49–108°). The study by Knott P. et al. [38] also included a group of patients with kyphosis.

Regarding the type of scoliosis, only eight [20,21,33,38,40,45,46] studies specified the curve type of scoliosis in subjects. One [21] of these studies restricted subjects to the curve type of scoliosis that was double spinal curve (convexity to the right in the thoracic region and convexity to the left in the lumbar region), and the curve type of scoliosis in the other seven studies was diverse. The other ten studies did not specify the curve type of scoliosis of the subjects.

#### 3.3.3. Data Acquisition

The data collected in the experiment included radiological data and surface topographic data. The acquisition methods were taking radiographs and scanning the surface topography, respectively. Of the eighteen studies included, X-rays were obtained during the experiments in nine studies [20,32,34,35,37,38,43,45,47]. In the remaining studies, some did not use X-rays, and others used X-rays obtained before the experiment. ST data acquisition for all studies was performed during the experiments.

### 3.4. Reliability Results

A total of eleven studies [20,32,33,37,38,39,41,42,44,46,47] underwent reliability assessment, and the results are presented in Table 6. The names (including abbreviations) and the definitions for all ST measurements are listed in Table 7. Among them, two studies [20,33] only evaluated the inter-investigator reliability, six studies [37,38,39,41,44,47] only evaluated the intra-investigator reliability, and three studies [32,42,46] evaluated both the inter- and intra-investigator reliability. 

Of the eleven studies for which reliability was assessed, nine [32,33,37,38,39,41,44,46,47] used the intra-class correlation coefficient (ICC) as the evaluation index, one study [20] used the Spearman correlation coefficient, and one study [42] used the variance coefficient (CV). Of the nine studies that used ICC, five [32,37,41,44,47] specified the ICC type or formula (listed in Table 4), one [45] did not specify the type but stated no systematic difference (*p*-value < 0.05), and four did not explicitly describe the details of ICC use. The study by Bolzinger M. et al. [20] and Mínguez M. F. et al. [42,43] did not report whether there were systematic differences. Some studies [33,44,47] also used the estimates of measurement errors, such as standard error of measurement (SEM) and smallest detectable change (SDC), and the specific values are extracted and listed in Table 6.

#### 3.4.1. Intra-Investigator Reliability Assessment Results

Nine studies [32,37,38,39,41,42,44,46,47] conducted intra-investigator reliability evaluations. Among them, eight studies [32,37,38,39,41,44,46,47] used the intra-class correlation coefficient as the evaluation index, and one study [42] used the coefficient of variation. 

In terms of automation level, six [32,38,42,44,46,47] studies manually identified or placed marker points, and three [37,39,41] studies used automatic identification. The intra-investigator reliability of measurements using both automatic and manual identification is good to excellent (evidence level: strong).

In most of these studies, the type of scan was back-only, with only two studies using a full-torso scan format. Four back-only studies used the Formetric system, and all measurements obtained good to excellent results (evidence level: strong). Moreover, the two studies that used full-torso scan achieved good to excellent results for their measurements (evidence level: moderate). 

To further summarize the results regarding different body planes and regions, we divided the ST measurements into four categories: Sagittal, horizontal, frontal, and asymmetry measurements. Asymmetry measurements were used in six studies [37,39,41,42,44,46], sagittal measurements in six studies [32,37,38,41,44,47], frontal measurements in five studies [32,37,38,41,44], and horizontal measurements in four studies [32,37,41,44]. Good to excellent intra-investigator reliability was shown on all asymmetry, sagittal, and horizontal measurements and on most frontal measurements (evidence level: strong). Only the pelvis obliquity (PO) parameter in the study by Tabard-Fougere et al. [32] showed poor results (evidence level: limited). However, pelvis-related parameters were also present in two other studies [41,44], which showed good to excellent results (evidence level: moderate).

#### 3.4.2. Inter-Investigator Reliability Assessment Results

Among the five studies [20,32,33,42,46] for which inter-investigator reliability was evaluated, three studies [32,33,46] used intra-class correlation coefficient (ICC), one study [20] used Spearman correlation coefficient (SCC), and one study [42] used coefficient of variation (CV) as the evaluation index. 

All five studies used the back-only scan format, and the systems used included self-designed systems [42,46], Formetric 4D [32], and BIOMOD^®^L [20,33]. The self-designed systems were similar; both included projectors and their measurements had good inter-investigator reliability (evidence level: moderate). Regarding automation level, four studies [32,33,42,46] manually identified or placed marker points, and only one study [20] used automatic identification. The inter-investigator reliability was good for measurement derived from automatic identification (evidence level: limited) and varied more for manual identification of reference landmarks. 

Regarding body parts and planes, three studies [20,42,46] used asymmetry measurements, and two studies [32,33] used sagittal, horizontal, and frontal measurements. The results showed that asymmetry ST measurements including axial plane deformity index (DAPI), horizontal plane deformity index (DHOPI), posterior trunk symmetry index (POTSI), and columnar profile (PC) had good to excellent inter-investigator reliability (evidence level: moderate). Most of the sagittal and frontal measurements had good to excellent reliability, with only some in the lumbar and pelvic regions showing decreased reliability. In particular, the pelvic region measurements showed poor results (evidence level: moderate). The inter-investigator reliability of horizontal measurements was good to excellent, with only one measurement (thoracic gibbosity) [33] showing poor reliability (evidence level: moderate).

### 3.5. Validity Results

A total of fourteen studies [21,32,34,35,36,37,38,39,40,42,43,45,46,47] were evaluated for validity, and the results are presented in Table 8. The names (including abbreviations) and the definitions for all ST measurements are listed in Table 7. Of these, thirteen studies used the Pearson correlation coefficient as an evaluation measure (only one [32] reported both r and *p*-values of *t*-tests), and one used the *p*-values of *t*-tests and linear regression.

Due to the variety of measurements, the results can be different when correlating a specific ST measurement with different radiological measurements. Therefore, we made the following classifications: (1) ST measurements and radiological measurements used for comparison are the same (for asymmetry measurements, the same radiological parameter is the Cobb angle); (2) ST measurements and radiological measurements used for comparison are different.

The validity results were satisfactory when using the same measurements for comparison. The results of four studies [32,34,35,38] showed that most frontal measurements (mainly scoliosis angles) have good to strong correlations (evidence level: strong). Similarly, in three studies [34,38,47], the sagittal measurements (mainly kyphosis angle and lordosis angle) showed good to strong correlations (evidence level: strong). There was one study exception [38] where two ST measurements (lumbar scoliosis curve and sagittal vertebral axis) showing moderate correlation occurred. Horizontal plane measurements (vertebral rotation) were addressed in only one study, indicating a moderate correlation (evidence level: limited). Regarding asymmetry measurements, six studies analyzed six measurements (PD [36], POTSI [42,45,46], DAPI [42], DHOPI [45,46], RMS [21], and asymmetry index [39,40]), mostly obtaining good to strong results (level of evidence: strong). However, moderate results appear in the calculation of POTSI with Cobb angle [46] and RMS with lumbar Cobb angle [21] (evidence level: limited). Five studies [32,34,38,43,47] in this category used the Formetric system (back-only type) and four of them showed good correlations between corresponding measurements (evidence level: strong). The validity of measurements derived from manual markers and automatic identification did not demonstrate a noticeable difference.

When different measurements were used for comparison, the validity results of the various measures appeared to differ. For example, two studies [45,46] comparing the relationship between PC and different radiological measurements obtained poor to good results (evidence level: moderate). Another study [37] correlated different ST measurements with radiological measurements and obtained moderate validity results (evidence level: limited).

## 4. Discussion

This systematic review revealed some results with strong evidence levels for both intra-investigator reliability and validity. Regarding intra-investigator reliability: (1) Both automatic and manual identification had good to excellent intra-investigator reliability. (2) The Formetric measurements obtained good to excellent results. (3) Good to excellent intra-investigator reliability was shown on all asymmetry, sagittal, and horizontal measurements and most frontal measurements. Regarding validity, when comparing the same ST and radiological measurements: (1) Most frontal, sagittal, and asymmetry measurements have good to strong results. (2) The Formetric measurements obtained good results. With regard to inter-investigator reliability, no results with a strong level of evidence were found due to the small number of studies (only five) and the differences in measurements used across studies. However, it was still found that the self-designed system (using projectors), asymmetry measurements, and most of the frontal and sagittal measurements (with the exception of lumbar and pelvic regions) had good to excellent inter-reliability at a moderate level of evidence.

The above results show that most ST measurements have satisfactory reliability and validity, especially for asymmetry, frontal, and sagittal measurements (with the exception of lumbar and pelvic regions). The Formetric measurements provide good intra-investigator reliability and validity. Nevertheless, the inter-investigator reliability results need to be further investigated (only one of the included studies addressed the inter-investigator reliability of the Formetric measurements). 

Regarding why lumbar and pelvic measurements (pelvis obliquity, pelvic imbalance, lumbar sinuosity) show poor intra- and inter-investigator reliability, we speculated that the reason might be related to the structure of the pelvic area and the artificial placement or selection of the landmarks. On the contrary to radiological methods that can irradiate directly through the skin and soft tissues to the internal skeletal structures of the body, surface topography methods determine the location of anatomical landmarks based on the surface morphology of the human body. Holcombe [50] et al. studied the distribution of subcutaneous fat in humans for each vertebral plane from T6 to the sacrum. They found that the average thickness of subcutaneous fat was greater in the areas on both sides of the lumbar spine (mainly in the L3–L5 vertebral plane) than in other areas of the back. This distribution characteristic leads to a larger distance from the spine to the skin of the back [51] and sometimes more skin folds in the lumbar region.

Moreover, to protect the subject’s privacy, the sacrococcygeal region is often covered by clothing during the test and cannot be fully exposed. Therefore, skin folds, fat content, and clothing obscuration may affect the accuracy of surface topography data measurement and calculation. There are limitations in measurement accuracy in obese patients, patients with asymmetrical muscle surfaces, and patients who have undergone surgery, as mentioned in the study by Asamoah et al. [52]. Among the studies included in our review, three studies [21,34,38] included both thoracic and lumbar segment indices in their results. The comparison revealed that the validity of the thoracic segment indices was better than the lumbar segment indices, which corroborates the influence of the factors mentioned above on the reliability and validity results. 

In addition, some ST systems require the operator to manually place the marker points at the specified anatomical locations to increase the accuracy of the measurement. In this case, it is not easy to accurately palpate or locate the marker when placing it due to the privacy of the sacrococcygeal region, and the position of the marker placed by different operators may vary, which may be another type of reason that affects the reliability. In the studies we reviewed, intra-investigator reliability results for manual marker identification were satisfactory, but inter-investigator reliability showed significant differences. Furthermore, this could ensure that operator differences in marker placement and selection affect reliability.

The methodological quality is essential when conducting reliability or validity evaluation. During our quality evaluation, some common methodological weaknesses were identified. First, many studies did not specify the qualifications of the testers, e.g., whether prior training had been conducted. In studies with automatic identification, the impact may not be significant, but for those cases where manual identification of anatomical markers is required, whether the researcher is sufficiently familiar with the location of the anatomical markers will undoubtedly impact the results. Some studies did not ensure that the testers were blinded to the subjects they have examined previously, which may exaggerate intra-investigator reliability. Some studies did not indicate whether the examination order varied, which may also affect the reliability. For example, testers may recall their previous measurements. In addition, possibly for radiation reduction, some studies used X-rays obtained before the experiment, which created a long time interval between ST and radiological measurements and may have affected the validity results. Therefore, the methodological quality can be improved by standardizing testers’ qualifications, using appropriate methods to ensure intra-investigator blinding, using a randomized test sequence, and shortening the time interval between ST and reference standard measurements.

By analyzing the studies [41,44] with better results in the lumbar-pelvic region, we found that automatic marker identification, prescribed subject standing position, and full-torso scan may be responsible for the improved outcomes. Compared with the manual identification of marker points, automatic identification can reduce the influence of operator competence factors on the results. The full-torso scan can provide three-dimensional information from multiple angles, somewhat compensating for the information obtained from the back-only scan. Specifying the subject’s standing position unifies the position and angle of all subjects relative to the device. It may allow the subject to be located at the optimal position and angle for device acquisition. In the analysis of the validity results, we found that in addition to the clarity of the bony markers of the lumbar region and the accuracy of the marker point placement, another possible reason for the poor validity was that the ST parameters and radiological parameters used to establish the correlation did not match. For example, the moderate correlation between rotation range and Cobb angle was shown in the study by Gorton [37] et al. The rotation range is a horizontal plane indicator, and the Cobb angle is a coronal plane indicator, both of which do not lie in the same plane. A similar situation exists in the study of Pino-Almero [45] et al. PC is a composite parameter consisting of three angles measuring the dorsal curvature in the sagittal plane (upper thoracic region, thoracolumbar region, and lumbosacral region, respectively). However, the radiological parameter with which the correlation was established was thoracic lordosis or lumbar lordosis, which is only one of PC’s components and therefore does not show a good correlation.

Of the four measurement types we analyzed, the asymmetry measurements are more promising than the other three categories, especially in the three-dimensional or comprehensive evaluation of scoliosis. The analysis showed that both intra- and inter-investigator reliability of asymmetric measurements was good, indicating their good reproducibility. Although asymmetry measurements do not yield the desired correlation when correlating with partial radiological measurements, as mentioned previously, this should not be a limitation for us. In practice, we should be more concerned with its reliability, which indicates whether the results are stable in repeated measurements and whether the results change when different testers perform the test process. Concerning validity, it is unnecessary to be overly concerned with the degree of match with existing radiological measurements. As Goldberg [35] mentioned, ST and radiological parameters measure different aspects of deformity. Rather than focusing significantly on the degree of matching ST measurements with radiological measurements, which limits the application of ST techniques, it is better to select better asymmetry measurements through research and establish corresponding evaluation criteria. In fact, this has been investigated by some researchers. For example, Mínguez [42] et al. used deformity in the axial plane index (DAPI) and posterior trunk symmetry index (POTSI) to establish a diagnostic criterion for scoliosis (DAPI ≤ 3.9% and POTSI ≤ 27.5%) and obtained good sensitivity and specificity. However, the ST technique has not established a gold standard for scoliosis diagnosis, such as the Cobb angle. Perhaps this is where future studies related to ST assessment of scoliosis can be focused. Since the Cobb angle is considered the gold standard in radiological measurements for the diagnosis of scoliosis and has an important place, there are many researchers and physicians who are interested in whether the ST parameter can replace the Cobb angle. Of the fourteen studies in which validity was evaluated, eleven compared the correlation between ST measurements and the Cobb angle. Good to strong correlations were obtained when comparing the ST scoliosis angle with the Cobb angle. Good correlations were also obtained for some asymmetry measurements and moderate or poor correlations for others when comparing the asymmetry measurements with the Cobb angle. The correlation is poor to moderate when comparing ST measurements that differed from the Cobb angle (e.g., rotation range). This indicates that the ST scoliosis angle and some asymmetry measurements can reflect the Cobb angle. However, ST measurements and Cobb angle do not match in other cases. Although the ST scoliosis angle and Cobb angle showed a high correlation in most cases in the relevant studies, the number of studies was limited (only four) and the spinal segments measured were not identical. Therefore, the evidence is not sufficiently strong to suggest that the ST scoliosis angle and Cobb angle can be used interchangeably in testing patients. More relevant studies, consistent results, and higher correlations are still needed to demonstrate that ST parameters and the Cobb angle can interchange.

In addition to being used to diagnose scoliosis, the Cobb angle is widely used by doctors to quantify the effectiveness of surgical or conservative treatments. However, on the contrary to doctors who focus on the internal Cobb angle, patients and their families are more concerned with improving the external shape or aesthetics. The expert consensus of the International Society on Scoliosis Orthopaedic and Rehabilitation Treatment (SOSORT) states that we should treat patients according to their future and current needs, where aesthetic issues should be the main reason for treating scoliosis [53]. However, the Cobb angle is not simply correlated with the external shape of the back. James [54] showed four girls with the same 70° Cobb angle in his study, with distinct individual differences in appearance. A study by Thulbourne et al. [55] also noted no clear linear relationship between rib deformity and the Cobb angle. For aesthetic issues, the value of the Cobb angle is not the only influencing factor; asymmetries in the shoulder, scapula, lumbar, and pelvic regions should also be taken into account [56]. Therefore, the surface topography technique may show advantages in evaluating the external shape. For example, asymmetry parameters can be combined with parameters from multiple planes or anatomical landmarks for a comprehensive analysis. The asymmetry parameter better reflects the shape of the back of the human body than traditional parameters (e.g., Cobb angle, thoracic kyphosis) that are analyzed based on a few anatomical landmarks. However, since radiological measurements mainly reflect the internal skeletal arrangement rather than the external shape, to evaluate the validity of ST measurements in reflecting external shape, more appropriate reference standards should be selected (such as the questionnaire on perceived appearance), which may provide more convincing evidence.

In addition to the emphasis on external shape improvement, progression identification is also essential during the follow-up of scoliosis. The need to monitor progress while avoiding the increased risk of cancer from repeated radiation exposures has led to the development of radiation-free methods, such as surface topography. Some studies have explored this issue. For example, Pino-Almero [43] et al. found a good correlation between the difference in DHOPI, POTSI, and Cobb angle in a 6-month interval. If these two parameters were used to follow the progression, a second radiography could be avoided in 70.96% (22 of 31 patients) cases. The result suggests that ST measurements also have the potential to identify the progression of scoliosis. However, there is still a need to demonstrate sensitivity to change and responsiveness to different parameters. Since this component was rarely addressed in the included studies, this point was not addressed in our evaluation and needed further investigation. Due to its non-radiation, non-invasive, and low-cost characteristics and its unique advantages in the three-dimensional evaluation and aesthetic assessment, the surface topography (ST) technique has great potential and promising applications in assessing scoliosis.

The results of this systematic review indicate that the ST technique has outstanding reliability (especially intra-observer reliability) when used for scoliosis assessment. Although some studies showed differences, the validity results were also satisfactory overall. Regarding reliability, accuracy can be improved by standardizing patient posture, calibrating anatomical landmarks, and using automated processing techniques [57]. Regarding validity, methods and standards within the ST technique should be established primarily rather than focusing only on improving its correlation with radiological parameters. The role of the ST technique should be a complement to the Cobb angle, not a substitute for it. Therefore, ST technology can be used in various areas, such as screening, outcome assessment, and follow-up of scoliosis and can provide a different perspective than radiology in assessing and treating scoliosis. Although the PRISMA guidelines were followed for the systematic evaluation, our review still has some limitations, which can be improved in further studies. Methodologically, when conducting literature screening and study quality assessment, we did not use the guidelines proposed by COSMIN, which may be more appropriate in evaluating measurement properties. In addition, some databases we did not use (such as CINAHL or SportDiscuss) may also be relevant to the topic. Moreover, we did not report on the agreement between reviewers during the inclusion or exclusion decision, data extraction, and methodological quality assessment phases. Furthermore, in the validity assessment, our criteria for interpreting correlation coefficients, although more commonly used, are relatively low (e.g., compared with those suggested by COSMIN) and may not be adequate to indicate that ST measurements could replace radiological measurements. At present, we are aware of COSMIN guidelines on conducting systematic reviews of measurements and that these documents could have guided the design of the search and the conducting of a more thorough quality appraisal. In the future, we will try to use the COSMIN guidelines to improve the quality of systematic reviews.

In addition, due to the limited number of included studies, some ST measurements were not adequately studied. Inter-investigator reliability for frontal, sagittal, and horizontal measurements did not obtain higher levels of evidence, especially for horizontal measurements, which showed inconsistent results. On the validity results of horizontal measurements, the evidence level was limited since only one study was involved. Studies on the reliability of horizontal measurements could be the focus in future studies. The inter-investigator reliability of automatic and manual identification methods also needs further investigation. For measurements from other common systems, such as InSpeck and Quantec, the evidence was not as conclusive.

Concerning the characteristics of the included studies, the subjects we included were all younger than 25 years of age. It is not possible to reflect whether different age groups may affect the reliability of ST measurements for evaluating scoliosis (e.g., older patients with degenerative scoliosis may have different results). With regard to the validity evaluation, we used a single reference standard. If other advantages of ST are considered, more diverse reference standards should be selected. Moreover, we did not review the sensitivity and responsiveness of ST measurements in identifying scoliosis progression. All of the above can be improved in future studies.

## 5. Conclusions

This study analyzed the reliability and validity of the surface topography technique for evaluating scoliosis through a systematic review of eighteen studies. The majority of the asymmetry, sagittal, and frontal ST measurements showed satisfactory reliability and validity. Horizontal ST measurements showed good reliability and poor validity. This review suggests that the ST technique may have great potential in assessing scoliosis, especially in reducing radiation exposure and performing cosmetic assessments.

## Figures and Tables

**Figure 1 jcm-11-06998-f001:**
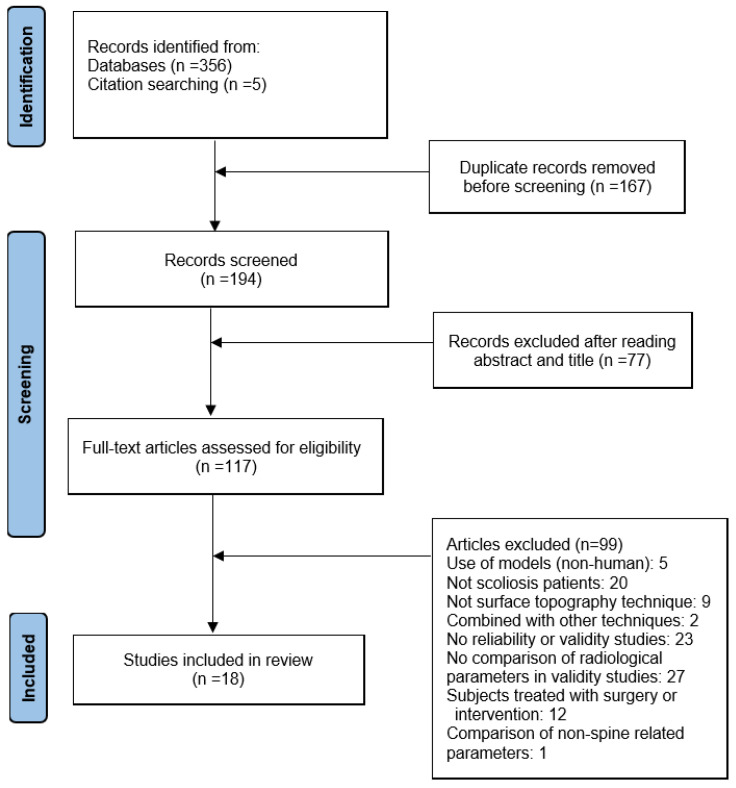
The flow chart of literature screening.

**Table 1 jcm-11-06998-t001:** Search strategy of PubMed.

#1	“Scoliosis” [MeSH] OR scoliosis OR scolioses OR “adolescent idiopathic scoliosis” OR “idiopathic scoliosis” OR “spinal curvature”
#2	“rasterstereography” OR “rasterstereographic” OR “surface topography” OR “stereophotogrammetry” OR “Moire topography” [MeSH]
#3	“Reproducibility of Results”[MeSH] OR “validity” OR “reliability” OR “validation” OR “accuracy” OR “validate”
#4	#1 AND #2 AND #3

**Table 2 jcm-11-06998-t002:** Level of evidence.

Strong	Consistent findings among ≥ three high-quality studies
Moderate	Consistent findings among ≥ one high-quality study and ≥ one low-quality study
Limited	Findings from ≥ one low-quality study or only from one study (high- or low-quality)
Conflicting	Inconsistent findings among ≥ two studies (high- or low-quality)

**Table 3 jcm-11-06998-t003:** Criteria for interpreting ICC, r, and CV values.

ICC Value	CV Value	Reliability Level	r-Value	Correlation Level
>0.9	≤10%	excellent	>0.8	Strong
0.75–0.9	10–20%	good	0.6–0.8	Good
0.5–0.75	20–30%	moderate	0.3–0.6	Moderate
<0.5	>30%	poor	<0.3	Poor

**Table 4 jcm-11-06998-t004:** Results of study quality assessment.

	1	2	3	4	5	6	7	8	9	10	11	12	13	HQ
Tabard-Fougere, A. et al. [32]	N	N	Y	N	Y	Y	Y	Y	Y	N	Y	Y	Y	√
Bolzinger M. et al. [20]	Y	N	n/a	Y	n/a	N	n/a	N	n/a	N	n/a	Y	N	×
de Sèze M. et al. [33]	N	N	n/a	Y	n/a	N	n/a	N	n/a	N	n/a	Y	Y	×
Frerich J. M. et al. [34]	Y	N	Y	n/a	n/a	n/a	Y	Y	Y	Y	Y	Y	Y	√
Goldberg C. J. et al. [35]	Y	N	Y	n/a	n/a	n/a	Y	n/a	Y	Y	Y	Y	Y	√
Gonzalez-Ruiz J. M. et al. [36]	Y	N	Y	n/a	n/a	n/a	N	n/a	Y	Y	Y	Y	Y	√
Gorton G. E. et al. [37]	Y	N	Y	n/a	N	N	N	Y	Y	Y	Y	Y	Y	√
Knott P. et al. [38]	Y	N	Y	n/a	N	n/a	Y	Y	Y	Y	Y	Y	N	√
Sudo H. et al. [39]	Y	N	Y	n/a	N	N	N	Y	Y	Y	N	Y	N	×
Kokabu T. et al. [40]	Y	Y	Y	n/a	n/a	n/a	N	n/a	Y	Y	Y	Y	Y	√
Manca, A. et al. [41]	Y	N	n/a	n/a	N	N	n/a	Y	n/a	Y	n/a	Y	Y	√
Mínguez M. F. et al. [42]	Y	Y	Y	Y	N	N	N	Y	Y	Y	N	Y	N	√
Mangone, M. et al. [43]	Y	N	Y	n/a	n/a	n/a	Y	n/a	Y	Y	Y	Y	Y	√
Pazos V. et al. [44]	N	N	n/a	n/a	N	N	n/a	Y	n/a	Y	n/a	N	Y	×
Pino-Almero L. et al. [45]	Y	N	Y	n/a	n/a	n/a	N	n/a	Y	Y	Y	Y	Y	√
Pino-Almero L. et al. [46]	Y	Y	Y	N	N	N	N	N	Y	Y	Y	Y	Y	√
Yıldırım Y. et al. [21]	Y	N	Y	n/a	n/a	n/a	N	n/a	Y	Y	Y	N	Y	√
Tabard-Fougere, A. et al. [47]	Y	N	Y	n/a	N	N	Y	Y	Y	Y	Y	Y	Y	√

Questions: 1. Detailed description of subjects; 2. Clarified the qualifications of testers; 3. Explained reference standard; 4. Inter-raters blinded; 5. Intra-rater blinded; 6. Varied examination order; 7. The appropriate period between the reference standard and index test (within 7 days); 8. The appropriate time interval between repeated measures (within 7 days); 9. Independent reference standard; 10. A detailed description of the test process; 11. A detailed description of the reference standard process; 12. Explanation of withdrawals; 13. Appropriate statistical methods. Y = yes; N = no; n / a = not applicable; HQ = high-quality.

**Table 5 jcm-11-06998-t005:** Characteristics of the study.

	Study	Participant Information				ST System	Acquisition Protocol
		Number and Sex		Age	Degree and Type of Scoliosis		
1	Tabard-Fougere, A. et al. [32]	35 (22 females and 13 males)		13.1 ± 2.0 years	Not mentioned	Formetric 4D (Diers International GmbH; Schlangenbad, Germany)	Each patient received a postero-anterior biplanar radiography and ST scans; Three repeated ST scans by two different operators (OP1 and OP2) on the same day, three more ST scans by the first operator (OP1) 1 week later; Subjects stood with their fists on their clavicles, elbow flexed, and the head looking forward; Manual marking of anatomical landmarks; Back-only scanning
2	Bolzinger M. et al. [20]	123 (111 females and 12 males)		10–13	Cobb angles: 10–40°; Type: 70 single curve (24 thoracic, 18 thoraco-lumbar, 28 lumbar), 53 double curve (double thoracic or thoracic-lumbar)	BIOMOD^®^L (AXS MEDICAL, Merignac, France)	Each patient received three acquisitions at 6-month intervals; Each acquisition consists of two ST scans (performed by two operators separately); Acquisition posture and Scan-Scan interval not specified; Automatic measurement and calculation; Back-only scanning
3	de Sèze M. et al. [33]	46 (40 females and 6 males)		/	Cobb angles: 26.8 ± 10°; Type: 9 single curve, 36 double curve, 1 triple curve	BIOMOD^®^L (AXS MEDICAL, Merignac, France)	Each patient received two acquisition series (performed by two operators separately); Each acquisition involved a preparation stage and three ST scans in three different positions (Backward standing with different arm positions); Scan-Scan interval not specified; Manual marking of anatomical landmarks; Back-only scanning
4	Frerich J. M. et al. [34]	RE	14 females	16–25	Cobb angles: 9 participants <10°, 5 participants 15–40°; Type: Not mentioned	Formetric (Diers Medical Systems, Chicago, IL, USA)	Each participant received thirty ST scans in a 60-min time period (subjects stood in their normal, comfortable posture); Automatic measurement and calculation; Back-only scanning
		VA	64 (55 females and 9 males)	9–17	Cobb angles: 10–50°; Type: Not mentioned		Each patient received a standard postero-anterior radiograph and thirty ST scans in a 60-min time period (subjects stood in their normal, comfortable posture); Automatic measurement and calculation
5	Goldberg C. J. et al. [35]	155 (132 females and 23 males)		2.67–20.58	Cobb angles: 41.16 ± 22°; Type: Not mentioned	Quantec (Quantec Image Processing, Warrington, Cheshire, UK)	Each subject received one spine radiograph and four ST scans; Subjects stood free in a customized frame, with feet separated by a standard wooden block; Scan-Scan interval not specified; Manual marking of anatomical landmarks; Back-only scanning
6	Gonzalez-Ruiz J. M. et al. [36]	21		<18	Cobb angles: 23.53 ± 9°; Type: Not mentioned	Self-designed system (Artec Eva MHT scanner, Viewbox 4.0 software)	Each subject received surface scan; Subjects used a standardized standing position with the arms raised and slightly flexed behind the head; Manual marking of anatomical landmarks; Full-torso scanning
7	Gorton G. E. et al. [37]	36 (26 females and 10 males)		10.8–17.7	Cobb angles: 49–108° Type: Not mentioned	Vitus Smart 3D Body Scanner (Vitronic, Wiesbaden, Germany)	Each subject received one spine radiograph and three ST scans; Subjects stood on a platform within a 1.6 × 1.8-m measurement volume, with hands at the side and arms slightly abducted; Scans were consecutive; Automatic measurement and calculation; Full-torso scanning
8	Knott P. et al. [38]	193 (148 females and 45 males)		8–18	thoracic average 22.7 ± 10°; lumbar average 19.6 ± 9°; kyphosis magnitude 54.0 ± 11°	Formetric (Diers Medical Systems, Chicago, IL, USA)	Each subject received standing postero-anterior and lateral radiograph and three ST scans; The ST scan was obtained three times within a 5-min period; Subjects stood in an upright position in front of the ST scanner; External markers are placed for obese patients; Back-only scanning
9	Sudo H. et al. [39]	76		7–18	Cobb angles: 0–64° Type: 59 single curve (33 thoracic, 17 thoraco-lumbar/lumbar), 19 double curve (thoracic and thoraco-lumbar/lumbar)	Self-designed system: A 3D depth sensor (Xtion Pro Live, ASUSTeK Computer Inc. Taipei, Republic of China), a computer	Forty-six subjects received one ST scan, thirty subjects received two ST scans with repositioning; Subjects bent forward below the sensor; Automatic measurement and calculation; Scan-Scan interval not specified; Back-only scanning
10	Kokabu T. et al. [40]	170 (149 females and 21 males)		8–18	Cobb angles: 0–60.7° Type: 119 single curve (70 thoracic, 49 thoraco-lumbar/lumbar), 47 double curve (thoracic and thoraco-lumbar/lumbar)	Self-designed system: A 3D depth sensor (Xtion Pro Live, ASUSTeK Computer Inc. Taipei, Republic of China), a computer	Each subject received one ST scan; Subjects bent forward below the sensor; Automatic measurement and calculation; back-only scanning
11	Manca, A. et al. [41]	66(53 females and 13 males)		10–17	Cobb angles: 22.9 ± 10.8° Type: 52 single curve (17 thoracic, 24 thoraco-lumbar, 11 lumbar), 14 double curve (thoracic and lumbar)	Formetric (Diers Medical Systems, Chicago, IL, USA)	Each subject received three ST scans; The assessment was repeated on the same day; Scan-Scan interval is 15 min (Retest 1) and 1 week (Retest 2); Subjects stood backward in free bipedal, heels placed at the end of the platform; Automatic measurement and calculation; Back-only scanning
12	Mínguez M. F. et al. [42]	30 (22 females and 8 males)		average: 14.88	Cobb angles: ≥10° Type: Not mentioned	Self-designed system: Screen, camera, computer, projector	Each subject received ST scan; twenty subjects received ST scans with repositioning; Subjects stood backward and arms straight down and facing straight ahead in a natural position; Manual marking of anatomical landmarks; No indication of when the X-rays were obtained; Back-only scanning
13	Mangone, M. et al. [43]	25 patients (14 females and 9 males)		14 ± 3	Cobb angles: 30 ± 9° Type: Not mentioned	Formetric 4D (Diers International GmbH, Schlangenbad, Germany)	Each subject received X-ray and ST scan; Scan-Scan interval not specified; Automatic measurement and calculation; Back-only scanning
14	Pazos V.et al. [44]	49 (42 females and 7 males)		11.0–19.7	Not mentioned	InSpeck system (InSpeck Inc., Montreal, Canada)	Each subject received four ST scans (two ST scans in two different arm postures); Subjects stood in the center of the system with the arms in slight abduction with the side or the elbows extending forward, hands on the side of the neck; Manual marking of anatomical landmarks; Half-min interval between two postures; Full-torso scanning
15	Pino-Almero L. et al. [45]	88 (76 females and 12 males)		7–17	Cobb angles: 10–51.80° Type: 39 single curve (8 thoracic, 17 thoraco-lumbar, 14 lumbar), 49 double curve (thoracic and lumbar)	Self-designed system (A mobile white screen, a projector, a digital camera, a computer with image recognition software)	Each subject received standard radiographies of the full spine and ST scan; Subjects stood backward with arms relaxed at the sides; Manual marking of anatomical landmarks; Back-only scanning
16	Pino-Almero L. et al. [46]	31 (27 females and 4 males)		7–17	Cobb angles: 13.10–35.00° Type: 39 single curve (8 thoracic, 17 thoraco-lumbar, 14 lumbar), 49 double curve (thoracic and lumbar)	Self-designed system (A mobile white screen, a projector EPSON, a digital camera Canon, and a computer with the program)	Each subject received two acquisitions at 6-month intervals; Each acquisition consists of two ST scans (performed by two operators separately) and a standard radiography of the entire spine; Subjects stood backward with arms hanging at the sides; Manual marking of anatomical landmarks; Scan-Scan interval not specified; Back-only scanning
17	Yıldırım Y. et al. [21]	42 (32 females and 10 males)		10–20	Cobb angles: Not mentionedType: Double curve (right convexity of thoracic and left convexity of lumbar)	A hand-held 3D scanner device (Artec EVA, Artec Group 2013, Luxembourg)	Each subject received ST scan in three different body positions (standing position with the arms hanging at the sides, standing position with the arms extended forward in a bending position); Manual marking of anatomical landmarks; Back-only scanning
18	Tabard-Fougere, A. et al. [47]	51 (32 females and 19 males)		13.5 ± 2.0 years	Cobb angles: 22.9 ± 17.4° Type: Not mentioned	Formetric 4D (Diers International GmbH, Schlangenbad, Ger-many)	Each subject received biplanar radiography and ST scan on the same day; Subjects stood with their fists on their clavicles, elbow flexed, and the head facing forward; Manual marking of anatomical landmarks; Back-only scanning

RE = reliability; VA = validity.

**Table 6 jcm-11-06998-t006:** Reliability results.

	Study	Type	Indicator	ST Measurements and Outcomes
1	Tabard-Fougere, A. et al. [32]	Intra	same day: ICC [1, 1]	[FM] SA (17.4 ± 7.1) ^1^: good (0.70); PO (3.0 ± 2.0): poor (0.50); [SM] TL (405.9 ± 36.6): excellent (0.97); TK (38.4 ± 11.0): excellent (0.94); LL (37.2 ± 9.7): good (0.86); [HM] VO_rms (5.6 ± 2.8): excellent (0.91); VO_max (10.0 ± 4.6): good (0.85); VO_amp (12.3 ± 6.4): good (0.88)
			1 week later: ICC [1, 3], SEM, SDC	[FM] SA (18.7 ± 5.5): moderate (0.72, 5.2, 4.4); PO (3.8 ± 3.4): poor (0.27, 3.7, 10.4); [SM] TL (406.6 ± 43.4): excellent (0.95, 13.1, 36.2); TK (34.8 ± 11.8): good (0.85, 6.1, 17.1); LL (35.6 ± 11.2): excellent (0.93, 3.8, 10.6); [HM] VO_rms (5.6 ± 2.6): good (0.88, 1.4, 3.9); VO_max (10.0 ± 4.4): excellent (0.90, 2.1, 5.9); VO_amp (12.8 ± 5.6): excellent (0.92, 2.8, 7.6)
		Inter	ICC [3, 3], SEM, SDC	[FM] SA (16.6 ± 5.9): good (0.84, 4.8, 3.4); PO (4.5 ± 3.6): poor (0.46, 3.4, 9.4); [SM] TL: excellent (0.96, 13.2, 36.6); TK: excellent (0.92, 6.1, 16.8); LL: excellent (0.94, 4.6, 12.7); [HM] VO_rms (5.6 ± 2.6): excellent (0.94, 1.2, 3.4); VO_max (9.7 ± 4.5): excellent (0.97, 1.5, 4.2); VO_amp (11.8 ± 6.2): excellent (0.94, 2.9, 8.0)
2	Bolzinger M. et al. [20]	Inter	SCC	[AM] RPC (202.7 ± 111, 200.8 ± 112) ^2^: good (0.8)
3	de Sèze M. et al. [33]	Inter	ICC, TEM	*“clavical” position* [FM] TS (11.5 ± 18.2) ^1^: excellent (3.8, 0.96); TLS (−8.3 ± 16.2): excellent (3.4, 0.96); LD_C7 (−5.9 ± 14.6): good (5.2, 0.88); PI (2.1 ± 2.4): moderate (1.3, 0.69); LS (−12.8 ± 13.8): poor (13.0, 0.13); [SM] AR_C7 (38.3 ± 20.2): good (7.1, 0.88); AR_T (7.3 ± 9.7): good (4.5, 0.79); HOIP (49.8 ± 10.8): good (3.8, 0.88); KA (29.9 ± 11.0): good (4.3, 0.85); AR_L (46.1 ± 9.7): moderate (6.0, 0.63); LA (36.0 ± 6.8): good (2.8, 0.83); [HM] TG (5.2 ± 2.7) ^1^: moderate (1.5, 0.70) ^2^; TH (74.3 ± 11.3): good (4.9, 0.82); PLP (4.7 ± 3.5): good (1.4, 0.84); LH (30.6 ± 7.2): good (3.2, 0.81); TLG (4.7 ± 3.1): good (1.5, 0.78); *“folding” position* [FM] TS (12.0 ± 19.1): excellent (3.7, 0.97), TLS (−8.6 ± 16.4): excellent (4.6, 0.93); LD_C7 (−4.0 ± 16.4): excellent (5.4, 0.90); PI (2.1 ± 2.5): moderate (1.6, 0.61); LS (−10.1 ± 14.2): moderate (9.7, 0.61); [SM] AR_C7 (33.3 ± 20.5): (9.3, 0.80); AR_T (4.1 ± 11.0): (5.0, 0.80); HOIP (48.7 ± 13.5): (4.5, 0.90); KA (32.1 ± 11.1): (4.3, 0.86): good to excellent; AR_L (4.1 ± 11.0): (6.2, 0.76); LA (50.4 ± 12.3): (2.7, 0.86): moderate to good; [HM] TG (5.6 ± 2.9): moderate (1.5, 0.74) ^2^; TH (76.3 ± 9.6): good (4.0, 0.84); PLP (5.2 ± 3.9): good (1.4, 0.87); LH (30.5 ± 6.8): good (2.4, 0.88); TLG (6.0 ± 3.3): moderate (1.8, 0.70) *“straight out” position* [FM] TS: (13.4 ± 18.0): excellent (4.2, 0.95); TLS (−9.2 ± 15.7): excellent (4.8, 0.91); LD_C7 (−5.8 ± 16.4): excellent (4.5, 0.93); PI (1.9 ± 2.4): moderate (1.7, 0.52); LS (−13.6 ± 8.6): poor (6.8, 0.44); [SM] AR_C7 (35.8 ± 19.4): excellent (6.1, 0.91); AR_T (4.5 ± 6.9): good (3.3, 0.78); HOIP (49.0 ± 12.5): good (5.8, 0.79); KA (31.5 ± 11.4): excellent (3.0, 0.93); AR_L (47.3 ± 10.8): good (5.3, 0.78); LA (37.1 ± 7.0): moderate (4.5, 0.60); [HM] TG (5.0 ± 2.9): moderate (1.9, 0.59); TH (75.0 ± 10.8): good (4.2, 0.86); PLP (5.3 ± 4.0): excellent (1.1, 0.93); LH (31.6 ± 7.3): excellent (2.0, 0.93); TLG (5.1 ± 3.2): good (1.4, 0.82)
4	Gorton G. E. et al. [37]	Intra	ICC [1, 3] (single measures) ICC [3, 3] (average measures)	[FM] MRS: excellent (0.94, 0.98); MLS: good to excellent (0.88, 0.96); SR_RL: excellent (0.90, 0.96); [SM] MPS: good to excellent (0.78, 0.91); MAS: good to excellent (0.87, 0.95); SR_AP: moderate to good (0.67, 0.86); [HM] MR_CCW: excellent (0.96, 0.99); MR_CW: excellent (0.97, 0.99); ROR: excellent (0.97, 0.99); [AM] SRE: excellent (0.98, 0.99); LRE: excellent (0.96, 0.99); RRE: excellent (0.94, 0.98); RLA_Min: good to excellent (0.75, 0.90); RLA_Max: moderate to good (0.63, 0.84); RLA_Ra: moderate to good (0.59, 0.81)
5	Knott P. et al. [38]	Intra	ICC	[FM] TC: excellent (0.95); LC: good (0.86); PO: good (0.894); T_IM: excellent (0.95) [SM] TK: excellent (0.98); LL: excellent (0.98); T_IN: excellent (0.91)
6	Sudo H. et al. [39]	Intra	ICC	[AM] AI: excellent (0.995)
7	Manca, A. et al. [41]	Intra	ICC [1, 2]	[FM] T_IM: good (0.88); PO: excellent (0.90); SD_amp: excellent (0.91); SD_rms: good (0.82); SA: excellent (0.98); [SM] T_IN: excellent (0.97); P_TI: excellent (0.93); TK: excellent (0.97); LL: excellent (0.97); [HM] P_TO: good (0.80); VO_rms: excellent (0.94); VO_amp: good (0.87)
8	Mínguez M. F. et al. [42]	Intra	CV	[AM] DAPI (0.27 ± 0.2) ^1^: excellent (5.13%); POTSI (3.95 ± 4.15): good (15.09%)
		Inter	CV	[AM] DAPI (0.38 ± 0.3): excellent (7.69%); POTSI (3.98 ± 4.17): good (15.16%)
9	Pazos V. et al. [44]	Intra	ICC, TEM, SDD	*“anatomical“position*: excellent. [FM] DCC (17.6 ± 9.8) ^1^: (3.19, 5.62, 0.91); LTI (0.4 ± 3.4): (0.88, 1.55, 0.97); FPT (0.2 ± 3.4): (0.69, 1.22, 0.97); [SM] STI (−5.6 ± 3.3): (0.75, 1.32, 0.95); [HM] BSR_max (11.6 ± 4.5): (1.41, 2.48, 0.93); ATR_max (6.8 ± 3.5): (0.84, 1.48, 0.97); TH (394 ± 30): (3.0, 5.3, 0.99); [AM] FAR (1.8 ± 1.6): (0.28, 0.59, 0.99); FA (13.2 ± 36.7): (4.74, 8.35, 0.99); LAS (5.4 ± 10.2): (2.66, 4.69, 0.95) *“clavicle” position:* good to excellent. [FM] DCC (19.7 ± 9.9): (3.90, 6, 87, 0.85); LTI (0.9 ± 3.4): (1.17, 2.06, 0.92); FPT (0.0 ± 3.4): (1.05, 1.85, 0.95); [SM] STI (0.2 ± 3.1): (1.54, 2.71, 0.88); [HM] BSR_max (12.0 ± 4.8): (1.36, 2.39, 0.92); ATR_max (6.8 ± 3.5): (0.81, 1.43, 0.97); TH (382 ± 31): (5.17, 9.11, 0.97); [AM] FAR (1.7 ± 1.5): (0.52, 0.92, 0.90); FA (12.9 ± 35.7): (4.89, 8.62, 0.98); LAS (5.0 ± 10.3): (2.48, 4.37, 0.94)
10	Pino-Almero L. et al. [46]	Intra	ICC	[AM] DHOPI: excellent (0.983); POTSI: excellent (0.959); PC: excellent (0.984) (*p* < 0.05)
		Inter	ICC	[AM] DHOPI: excellent (0.987); POTSI: excellent (0.978); PC: excellent (0.969) (*p* < 0.05)
11	Tabard-Fougere, A. et al. [47]	Intra	ICC [1, 1]	[SM] TK (36.9 ± 11.5) ^1^: excellent (0.937); LL (36.5 ± 9.9): excellent (0.965)

Inter = Inter-investigator reliability; Intra = Intra-investigator reliability; ICC = Intra-class correlation coefficient; SEM = Standard Error of Measurement; SDC = Smallest Detectable Change; TEM = Typical-Error Measurement; SCC = Spearman correlation coefficient; CV = Coefficient of variation; P = The *p*-value of a paired *t*-test; FM = Frontal measurements; SM = Sagittal measurements; HM = Horizontal measurements; AM = Asymmetry measurements; (frontal, sagittal, and horizontal measurements are those where the anatomical landmarks used in the measurement are located in the corresponding planes. Asymmetry measurements reflect body asymmetry and usually involve more than one plane or more than one body region). SA = Scoliosis angle; PO = Pelvis obliquity; TL = Trunk length; TK = Thoracic kyphosis angle; LL = Lumbar lordosis angle; VO_rms = RMS surface rotation T4-DM; VO_max = Maximal surface rotation T4-DM; VO_amp = Amplitude of surface rotation T4-DM; RPC = Rib prominence curve; TS = Thoracic sinuosity; TLS = Thoracolumbar sinuosity; LD_C7 = Lateral deviation of C7; PI = Pelvic imbalance; LS = Lumbar sinuosity; AR_C7 = C7 arrow; AR_T = Thoracic arrow; HOIP = Height of inflection point; KA = Kyphotic angle; AR_L = Lumbar arrow; LA = Lordotic angle; TG = Thoracic gibbosity; TH = Thoracic height; PLP = Paraspinal lumbar prominence; LH = Lumbar height; TLG = Thoracolumbar gibbosity; MRS = Maximum right shift; MLS = Maximum left shift; SR_RL = Right/left range; MPS = Maximum posterior shift; MAS = Maximum anterior shift; SR_AP = Anterior/posterior range; MR_CCW = Maximum CCW rotation; MR_CW = Maximum CW rotation; ROR = Rotation range; SRE = Smallest residual; LRE = Largest residual; RRE = Residual range; RLA_Min = Minimum right/left asymmetry; RLA_Max = Maximum right/left asymmetry; RLA_Ra = Right/left asymmetry range; TC = Thoracic curve; LC = Lumbar curve; T_IM = Trunk imbalance; T_IN = Trunk inclination; AI = asymmetry index; SD_amp = Side deviation-amp; SD_rms = Side deviation from symmetry line-rms; P_TI = Pelvis tilt; P_TO = Pelvis torsion; DAPI = Axial plane deformity index; POTSI = Posterior Trunk Symmetry Index; DCC = Maximum center Deviation; LTI = Lateral trunk inclination; FPT = Frontal pelvic tilt; STI = Forward-backward trunk inclination; BSR_max = Back surface rotation max; ATR_max = Axial trunk rotation max; TH = Trunk height; FAR = Frontal asymmetry ratio; FA = Frontal asymmetry; LAS = Lateral asymmetry; DHOPI = Axial plane deformity index; PC = Columnar Profile; ^1^ (Mean ± SD). ^2^ (Mean ± SD_OP1_, Mean ± SD_OP2_).

**Table 7 jcm-11-06998-t007:** Description of ST measurements.

Abbreviation	ST Measurement	Description	CL
AI (mm)	Asymmetry Index [39,40]	The asymmetry index is calculated by a series of complex steps including capturing, segmenting, estimating a median sagittal plane and the boundary, generating the reflection point cloud, fitting, and extracting deviations for the dorsal point cloud.	AM
AR_C7 (mm)	C7 plumb line arrow [33]	The horizontal anterior-posterior distance from the C7 spinous processes relative to the vertical line spanning the most posterior point of those processes.	SM
AR_L (mm)	Lumbar arrow [33]	The horizontal anterior-posterior distance between the most anterior point of the spinous processes at the level of the lumbar lordosis relative to the vertical line spanning the most posterior segment of the processes.	SM
AR_T (mm)	Thoracic arrow [33]	The horizontal anterior-posterior distance between the most posterior point of the spinous processes at the level of the thoracic kyphosis relative to the vertical line spanning the most posterior segment of the processes.	SM
ATR_max (°)	Axial trunk rotation max [44]	The maximum angle between the principal axis of the section and the X-axis ^1^.	HM
BSR_max (°)	Back surface rotation max [44]	The maximum angle between the tangent line to the back profile and the X-axis ^1^.	HM
DAPI	Axial plane deformity index [42]	Consists of an addition of the difference of the depths of symmetrical points, at the level of the scapulae and waist.	AM
DCC (mm)	Maximum center deviation [44]	The maximum deviation of the line passing through the centers of the first and last cross-sections ^2^ as a distance in medio lateral direction in the transverse plane.	FM
DHOPI	Horizontal Plane Deformity Index [45,46]	Two lines are drawn: (a) The line between the two most prominent points of the scapulae; (b) the line between the two least prominent points of the waist. Then, locating the symmetrical point of the most prominent point situated on the two lines. Finally, the differences in depth between the symmetrical points, divided by distance I ^3^, are added.	AM
FA (mm)	Frontal asymmetry [44]	The global apparent asymmetry in the frontal view was calculated as the difference between right and left areas.	AM
FAR	Frontal asymmetry ratio [44]	The global apparent asymmetry in the frontal view was calculated as the ratio.	AM
FPT (°)	Frontal pelvic tilt [44]	The angle between the horizontal plane and the ASIS line in the frontal plane.	FM
HOIP (%)	Height of inflection point [33]	The height of the intersection point by the spinous processes and the segment consisting in C7 and the intergluteal cleft (0% corresponding to the top of the intergluteal cleft and 100% to the C7 spinal process).	SM
KA (°)	Kyphotic angle [33]	The angle formed by the perpendicular lines at the C7 spinal processes and at the inflection point in the sagittal plane.	SM
LA (°)	Lordotic angle [33]	The angle formed by the perpendicular lines at the spinal processes at the inflection point and at the top of the intergluteal cleft in the sagittal plane.	SM
LAS (mm)	Lateral asymmetry [44]	In the lateral view, the global asymmetry was the root-mean-square difference between the right and left mid-lateral curves.	AM
LC (°)	Lumbar curve [34,38]	The angle of lumbar curve in the frontal plane between tangents to the cranial and caudal endplates of the respectively calculated cranial and caudal vertebral bodies.	FM
LD_C7 (mm)	The C7 lateral deviation [33]	The horizontal distance between the C7 spinous processes and the vertical plane spanning the top of the intergluteal cleft (positive on the right, negative on the left).	FM
LH (%)	Lumbar height [33]	Height of the maximum rotation in the lumbar deformity relative to the C7 intergluteal cleft.	HM
LL (°)	Lumbar lordosis angle [32,38,41,47]	Maximal lumbar angle calculated between the 12th thoracic vertebra (T12) and midpoint between DM in the sagittal plane.	SM
$\mathrm{LRE}\;(\sqrt{{\mathrm{mm}}^{2}}$)	Largest residual [37]	The largest root-mean-square error (RMSE) of the distance between the point cloud of any ellipse and the point cloud in the reference ellipse ^4^.	AM
LS (°)	Lumbar sinuosity [33]	The camber angles of the spinous processes situated between two inflection points of the line in lumbar area.	FM
LTI (°)	Lateral trunk inclination [44]	The shift of the VP relative to the vertical line passing through origin ^5^ in the frontal plane.	FM
MAS (mm)	Maximum anterior shift [37]	The maximum deviation from the reference ellipse ^1^ in anterior direction of any slice.	SM
MLS (mm)	Maximum left shift [37]	The maximum deviation from the reference ellipse ^1^ in left direction of any slice.	FM
MPS (mm)	Maximum posterior shift [37]	The maximum deviation from the reference ellipse ^1^ in posterior direction of any slice within the defined volume. The scan produced a number of slices. Each consists of several hundred data points (a “point cloud”) that are fitted to an ellipse. The reference ellipse was an ellipse fitted to the point cloud of the reference slice defined at the level of the posterior superior iliac spine. The center of this ellipse and orientation of its principal axis served as reference for all measurements.	SM
MR_CCW (°)	Maximum CCW rotation [37]	The maximum deviation from the reference ellipse ^1^ in counterclockwise (CCW) direction of any slice.	HM
MR_CW (°)	Maximum CW rotation [37]	The maximum deviation from the reference ellipse^1^ in clockwise (CW) direction of any slice.	HM
MRS (mm)	Maximum right shift [37]	The maximum deviation from the reference ellipse ^1^ in right direction of any slice.	FM
PC	Columnar Profile [45,46]	It is obtained by determining the three angles that are formed when identifying the following points in the topography: The first angle (PC1) is delimited by the line between the basis of the neck (C7 vertebra) with the inter-shoulder blade zone (T5 vertebra) and the vertical line. The second angle (PC2) is delimited by the line between the anterior point (T5) with the waist zone (L3) and the vertical line. The third angle (PC3) is delimited by the line between the previous point corresponding to L3 with the intergluteal cleft (sacrum) and the vertical line.	AM
PD (mm^2^)	Procrustes distance [36]	The sum of squared distances between landmarks and semi-landmarks after general Procrustes analysis (GPA) and allows for quantifying the differences in shape excluding size.	AM
PI (°)	Pelvic imbalance [33]	The angle formed between the horizontal plane and the line spanning the sacral dimples.	FM
PLP (°)	Paraspinal lumbar prominence [33]	The maximum rotation (relative to the frontal plane) of the paraspinal lumbar prominence.	HM
PO (mm)	Pelvis Obliquity [32,38,41]	Height difference between the right and left lumbar dimples, based on a horizontal plane.	FM
P_TI (°)	Pelvis tilt [41,44]	Angle between plumb line and a tangent on the lumbar dimples in the sagittal plane.	SM
P_TO (°)	Pelvis torsion [41]	Torsion between left and right side pelvis bones (os ilium).	HM
POTSI	Posterior Trunk Symmetry Index [42,45,46]	POTSI is the sum of two variables: Height asymmetry indices (HAI) and frontal asymmetry indices (FAI). HAI is obtained as the sum of height differences of the shoulders, axillary fold, and waist creases and is normalized with the division of its value by the distance I. FAI is the sum of the differences in horizontal distance with respect to the gluteal cleft, C7 vertebra, axillary folds, and waist, which is also normalized by dividing them by distance I ^3^.	AM
QA (°)	Quantec angle [35]	Derived from the spine line (defined by T1, T12, and the posterior superior iliac spines) in the frontal plane and is calculated automatically by the Quantec system.	FM
RLA_Max (mm^2^)	Maximum right/left asymmetry [37]	The maximum value of the difference between the area of the left and right halves (divided along the minor axis of the ellipse) of the point cloud of any ellipse.	AM
RLA_Min (mm^2^)	Minimum right/left asymmetry [37]	The minimum value of the difference between the area of the left and right halves (divided along the minor axis of the ellipse) of the point cloud of any ellipse.	AM
RLA_Ra (mm^2^)	Right/left asymmetry range [37]	The value range of the difference between the area of the left and right halves (divided along the minor axis of the ellipse) of the point cloud of any ellipse.	AM
$\mathrm{RMS}\;(\sqrt{{\mathrm{mm}}^{2}}$)	Root-mean-square [21]	Root-mean-square of the point-to-point distance after superimposing the original image and the mirror image.	AM
ROR (°)	Rotation range [37]	The deviation range from the reference ellipse ^1^ in CCW-CW direction of any slice.	HM
RPC (G)	Rib prominence curve [20]	The spine (between the spinous process of C7 and the top of the gluteal fold) is divided into 100 axial slices at equal distances. On each slice, a straight line is defined passing through the highest points of the back on either side of the midline. The angle between this line and the projection of the line passing through the two-posterior superior iliac spines is defined as the rib prominence angle (°). The height (%) of this slice is expressed using the percentage of spine height. The rib prominence curve is the sum of the rib prominence angles of all the slices included in the curve multiplied by the height value of the slice. The thoracic, thoracolumbar, and lumbar segment curves were calculated separately, and the largest of these was obtained as the final value of the rib protrusion curve.	AM
$\mathrm{RRE}\;(\sqrt{{\mathrm{mm}}^{2}}$)	Residual range [37]	The root-mean-square error (RMSE) range of the distance between the point cloud of any ellipse and the point cloud in the reference ellipse ^4^.	AM
SA (°)	Scoliosis angle [32,41]	Maximal angle in the frontal plane between tangents to the cranial and caudal endplates of the respectively calculated cranial and caudal vertebral bodies.	FM
SD_amp (mm)	Side deviation-amp [41]	Lateral deviations of vertebral bodies from symmetry line in the frontal plane as the maximal variation from VP to DM.	FM
SD_rms (mm)	Side deviation from symmetry line-rms [41]	Lateral deviations of vertebral bodies from symmetry line in the frontal plane as the central tendency from VP to DM.	FM
SR_AP (mm)	Anterior/posterior shift range [37]	The deviation range from the reference ellipse ^4^ in anterior-posterior direction of any slice.	SM
$\mathrm{SRE}\;(\sqrt{{\mathrm{mm}}^{2}}$)	Smallest residual [37]	The smallest root-mean-square error (RMSE) of the distance between the point cloud of any ellipse and the point cloud in the reference ellipse ^1^.	AM
SR_RL (mm)	Right/left shift range [37]	The deviation range from the reference ellipse ^4^ in right-left direction of any slice.	FM
STI (°)	Forward-backward trunk inclination [44]	The shift of the VP relative to the vertical line passing through origin ^3^ in the sagittal plane.	SM
TC (°)	Thoracic curve [34,38]	The angle of thoracic curve in the frontal plane between tangents to the cranial and caudal endplates of the respectively calculated cranial and caudal vertebral bodies.	FM
TG (°)	Thoracic gibbosity [33]	The maximum rotation (relative to the frontal plane) of the thoracic deformity.	HM
TH (°)	Thoracic height [33]	Height of the maximum rotation in the thoracic deformity relative to the C7 intergluteal cleft.	HM
T_IN (mm)	Trunk inclination [38,41]	Plumb line deviation distance from VP to DM in the sagittal plane. Sva = lateral trunk inclination.	SM
T_IM (mm)	Trunk imbalance [38,41]	Plumb line deviation distance from VP to DM in the frontal plane. Cva = forward-backward trunk inclination.	FM
TK (°)	Thoracic kyphosis angle [33,38,41,47]	Maximal thoracic angle calculated between VP and the 12th thoracic vertebra (T12) in the sagittal plane.	SM
TL (mm)	Trunk length [32,44]	Distance from VP to midpoint between DM.	SM
TLG (°)	Thoracolumbar gibbosity [33]	The maximum rotation (relative to the frontal plane) of the thoracolumbar deformity.	HM
TLH (%)	Thoracolumbar height [33]	Height of the maximum rotation in the thoracolumbar deformity relative to the C7 intergluteal cleft.	HM
TLS (°)	Thoracolumbar sinuosity [33]	The camber angles of the spinous processes situated between two inflection points of the line in thoracolumbar area.	FM
TS (°)	Thoracic sinuosity [33]	The camber angles of the spinous processes situated between two inflection points of the line in thoracic area.	FM
VR (°)	Vertebral rotation [43]	The angle between the surface orientation on spinous process line (the so-called symmetry line) and the normal to the frontal plane of the reference system.	HM
VO_amp (°)	Amplitude of surface rotation T4-DM [32,41]	Amplitude of vertebral rotation measured perpendicular to back surface over the processus spinosus as the central tendency from VP to DM.	HM
VO_max (°)	Maximal surface rotation T4-DM [32]	Absolute value of maximal vertebral rotation measured perpendicular to back surface over the processus spinosus as the central tendency from VP to DM.	HM
VO_rms (°)	RMS surface rotation T4-DM [32,41]	Root-mean-square of vertebral rotation measured perpendicular to back surface over the processus spinosus as the central tendency from VP to DM.	HM

CL = Classification; FM = Frontal measurements; SM = Sagittal measurements; HM = Horizontal measurements; AM = Asymmetry measurements; (frontal, sagittal, and horizontal measurements are those where the anatomical landmarks used in the measurement are located in the corresponding planes. Asymmetry measurements reflect body asymmetry and usually involve more than one plane or more than one body region). G = Gibbosity, indicates percentage degree; VP = vertebra prominent; DM = lumbar dimples; ^1^ The X-axis was the horizontal parallel to the ASIS line toward the right of the patient. ^2^ Two hundred and fifty horizontal cross-sections were generated between the PSIS level and the T1 level, which represented a vertical step of about 2.5 to 3 mm according to the trunk length. The cross-sections were approximated with cubic splines. ^3^ Distance I is the vertical distance from the C7 vertebra to the baseline of the gluteal cleft. ^4^ The scan produced a number of slices. Each consists of several hundred data points (a “point cloud”) that are fitted to an ellipse. The reference ellipse was an ellipse fitted to the point cloud of the reference slice defined at the level of the posterior superior iliac spine. The center of this ellipse and orientation of its principal axis served as reference for all measurements. ^5^ The origin was the normal projection of ASIS line middle-point on the back surface.

**Table 8 jcm-11-06998-t008:** Validity results.

	Study	Indicator	ST Measurements	Radiographic Measurements	Outcomes
1	Tabard-Fougere, A. et al. [32]	r and p	SA (16.0 ± 5.9°)	Cobb angle (16.6 ± 9.3°)	Good (r = 0.70) and were non-significantly different (*p* = 0.60)
2	Frerich J. M.et al. [34]	r	TC, LC, LL, TK	Same as the ST measurements	TC: strong (r = 0.87); LL: strong (r = 0.81); TK: good (r = 0.80); LC: good (r = 0.76)
3	Goldberg C. J. et al. [35]	r	QA (24.2 ± 14.0°)	Cobb angle (41.16 ± 22.0°)	Strong (r = 0.81)
4	Gonzalez-Ruiz J. M. et al. [36]	p, Linear regressions	PD	Cobb angle	Significant correlation between Cobb angle and PD (r = 0.38; *p* = 0.01)
5	Gorton G. E. et al. [37]	r	ROR, MPS, SR_AP, SRE, LRE	Cobb angle, Kyphosis magnitude, Lordosis magnitude	ROR with Cobb angle: moderate (r = 0.48); MPS with kyphosis magnitude: moderate (r = 0.51); SR_AP with Kyphosis magnitude: moderate (r = 0.43); SRE with Lordosis magnitude: moderate (r = 0.45); LRE with Lordosis magnitude: moderate (r = 0.44)
6	Knott P. et al. [38]	r	TK, LL, TC, LC, T_IM, T_IN, PO	Same as the ST measurements	TK: strong (r = 0.87); LL: strong (r = 0.82); TC: good (r = 0.73); T_IM: good (r = 0.62); LC: moderate (r = 0.49); T_IN: moderate (r = 0.49)
7	Sudo H. et al. [39]	r	AI	Cobb angle	Strong (r = 0.88)
8	Kokabu T. et al. [40]	r	AI	Cobb angle	Strong (r = 0.85)
9	Mínguez M. F. et al. [42]	r	DAPI, POTSI	Cobb angle, Vertebral rotation angle	DAPI with Cobb angle: good (r = 0.71); POTSI with Cobb angle: good (r = 0.67); DAPI with Vertebral rotation angle: good (r = 0.62); POTSI with Vertebral rotation angle: moderate (r = 0.52)
10	Mangone, M. et al. [43]	r	VR (4.99 ± 3.50°)	Vertebral rotation (9.93 ± 5.38°)	Moderate (r = 0.52)
11	Pino-Almero L. et al. [45]	r	DHOPI, POTSI, PC	Cobb angle, Thoracic kyphosis angle, Lumbar lordosis angle, Vertebral rotation	DHOPI with: Cobb angle: strong (r = 0.81); POTSI with Cobb angle: good (r = 0.63); PC with thoracic kyphosis angle: moderate (r = 0.45); poor: PC with lordosis lumbar angle (r = 0.26); DHOPI with vertebral rotation poor (r = 0.31); POTSI with vertebral rotation: (r = 0.32)
12	Pino-Almero L. et al. [46]	r	DHOPI, POTSI, PC	Cobb angle, Thoracic kyphosis angle	DHOPI with Cobb angle: good (r = 0.77, 0.77) ^1^; POTSI with Cobb angle: moderate (r = 0.54, 0.54); PC with Thoracic kyphosis angle: moderate (r = 0.53, 0.61)
13	Yıldırım Y. et al. [21]	r	RMS	Cobb angle	RMS with in the thoracic region: good(r = 0.80, 0.76, 0.71) ^2^; RMS with Cobb angle in the lumbar region: moderate to good (r = 0.56, 0.65, 0.63)
14	Tabard-Fougere, A. et al. [47]	r	TK (36.9 ± 11.5°), LL (36.5 ± 9.9°)	Thoracic kyphosis (34.4 ± 11.2°), Lumbar lordosis (60.4 ± 10.7°)	Good: TK (r = 0.737); LL (r = 0.625)

*p* = The *p*-value of a paired *t*-test; r = Pearson correlation coefficient; SA = Scoliosis angle; TC = Thoracic curve; LC = Lumbar curve; LL = Lumbar lordosis angle; TK = Thoracic kyphosis angle; QA = Quantec angle; PD = Procrustes distance; ROR = Rotation range; MPS = Maximum posterior shift; SR_AP = Anterior/posterior shift range; SRE = Smallest residual; LRE = Largest residual; T_IM = Trunk imbalance; T_IN=Trunk inclination; PO = Pelvis Obliquity; AI = Asymmetry Index; DAPI = Axial plane deformity index; POTSI = Posterior trunk symmetry index; VR = Vertebral rotation; DHOPI = Horizontal plane deformity index; PC = Columnar profile; RMS = Root-mean-square; ^1^ The two values respectively indicate the correlation between the ST and radiographic measurements obtained from the first and second acquisition; ^2^ The three values in order represent the correlation in the standing position with the arms hanging at the sides, with the arms extended, and in a forward bending position.

## Data Availability

Not applicable.

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
