# Peer review of "Reliability and Validity of Scoliosis Measurements Obtained with Surface Topography Techniques: A Systematic Review"

_jcm, 2022, doi:10.3390/jcm11236998_

Round 1

Reviewer 1 Report

The paper is very well documented and the analyses are pertinent.

I think, if it is possible, to mention in the article:

- from what age have the subjects scoliosis ?

- the age has an influence ?

Author Response

Date: Nov 10, 2022

Dear reviewer,

Thank you for your comments concerning our manuscript entitled” Reliability and validity of scoliosis assessment using surface topography techniques: A systematic review”(Manuscript ID: jcm-1951687). Those comments are all valuable and very helpful for revising and improving our paper and the essential guidance for our further research. We have studied the comments carefully and made corrections, which we hope meet with approval. Revisions to the manuscript were marked up using the “Track Changes” function, and the detailed responses to your comments are uploaded as a pdf attachment.

Special thanks to you for your good comments.

Sincerely,

Ye Liu

[email protected]

School of Sport Science

Beijing Sport University, Beijing, China

Reviewer 2 Report

JCM-1951687 my review

Title. I suggest replacing “assessment” by ‘measurements obtained with’ as it is better to specify the measurements properties of a specific measurement in a specific context than to ascribe those to a type of assessment method.

Abstract Similarly I suggest rephrasing the objective as:

 This study aimed to systematically review the reliability and validity of the surface topography measurements used for assessing scoliosis.

Line 12 replace assessment by measurement

L14 can you specify how many measurements were inventories with some reliability or validity data in those studies.

L17 It is incorrect to refer to a study having a certain level or reliability. You have to refer to measurements.

Results are vague with not indications or how many measurements meeting a specific standard.

The conclusion should not be about whether Surface topography (ST) had reliability and validity. I would recommend that you mention ST measurement and could break that down further in terms of landmark based or automated without reliance on landmarks or reflecting postural changes in different planes. I suspect some are back only and some are full-torso parameters as well. All those grouping a measurements could be examined with regards to meeting clear standards in terms of reliability and validity.

L 21 ST does not have the potential to treat scoliosis. I would clarify that it can inform treatment decisions but It is not a treatment.

Keywords. The MESH term for reliability is reproducibility of results. Could mention posture in addition or instead of assessment which is quite vague.

L41 Always spell Cobb with a capital C

L46-47 Avoid Scoliois patient and instead use patient first language as recommended by WHO “female patients with scoliosis” here

L 59 I suggest to replace the organized assessment by “a systematic review”

L61 replace techniques by measurements.

The intro could benefit from introducing different categories of measurements you wish to examined; EG back-only vs full torso measurements, Landmark-based or derived without relying on landmark digitization, reflecting a 3D change, or frontal, or sagittal or transverse plane postural change.

L65 it would be relevant to follow the methodology proposed by COSMIN for such a review. You may want to announce it here how it informed your approach.

L66 Why did you use both Pubmed and Medline. (Medline should be fully included in your Pubmed search). You may want to mention as a limitation that you did not use CINAHL or Sport discuss which may be relevant to this topic.

L68 The COSMIN group has proposed a filter that could be used to track reliability and validity studies in systematic review of these properties. (See COSMIN.nl) you may want to acknowledge not using it as a limitation.

Table 1. Your search strategy is overly simple for such a topic and likely would suffer both from not being sensitive enough and not being specific enough. You do not appear to have made use of indexed terms such as MESH terms for Pubmed and MEDLINE. It appears you did not use any synonyms of surface topography. In particular novel techniques may be referred as stereophotogrammetry and older and current approaches still use Moire topography techniques. (for example adding those terms bumps the results up on PUBMED from 160 to 351) Some papers may only refer to the manufacturer names of the surface topography scanners. You did not use any terms to limit your search to your reliability and validity focus While this may still allow you to find what you need, it would force you to screen a much larger number of references.

Did you use any search limitations. It does not appear you limited to humans, English or specific age groups for example.

L72 to 87 should be deleted as they were likely instructions from the journal template.

L90 justify you including all types of scoliosis. For example reliability may be quite different between cases of neuromuscular scoliosis with difficulty sitting / standing.

L93 refer to measurement rather than then technique.

L94. Surface topography is an external measurement which radiographs provided information about the internal skeletal alignment changes. Your intro could justify your choice of considering radiograph a reference standard for validation or your discussion should clearly expand on this important limitation.

L98 99 I suggest to clarify if you excluded all treatments. Otherwise explain how you determined whether treatment may have affected results and decided to exclude some but not all studies with treatments. Specify if patients having had surgery could be included in your review?

L106 Report whether you tracked reviewer agreement.

L107 to 112 there should be extraction of the measurements and methods used to obtained reference measurements used in validity analysis too.

L115. Ok. Nowadays the COSMIN tool is preferred in reviews on measurement properties. Consider mentioning in the discussion.

You need to outline a method for data synthesis.

For a narrative review explain how you combined findings from different studies on similar measurement and measurement properties to arrive at a single summary statement.

Specific which reliability and validity statistics you deemed adequate and outline your interpretation guides.

Outline your rules to consider any meta-analysis.

Explain how you examined heterogeneity qualitatively and / or quantitatively.

Outline which subgroupings of results you were interested in. For example see my notes above on different options.

Validity and reliability are broad categories of measurement properties, it would be relevant for the methods to announce which specific types under these heading you were interested in and what you wanted to summarize and what you considered adequate statistics under each.

Figure 1. You need to list the main exclusion reasons in the last box on the right.

L137 Refer to being blinded rather than blindness.

L137 In the method please outline what we deemed an appropriate period between measurements in the present study (between repeated reliability measurements and between index and standard tests).

Table 2. Many of the reference to the studies mention the author first names rather than the more typical format of Lastname, First initial.

L144 Typo to high quality.

L146 to 149 this content is redundant with the methods. Could you highlight some key findings? Or just refer to table 3.

Table 1 Ideally in the scanner can you find the type it takes to obtain a scan. It may be quite relevant to interpret the reliability results.

Patient positions should be described and whether a position support was used should be mentioned too.

Under acquisition protocol always specify the time between repeated scans. If not available please state interval between scan not available.
Where the type is know specify the percent of which diagnosis making up the samples.

L190 to 195 For the studies using ICC, can you report how many reported their ICC type and which types were used among those using it.

Cronbach alpha does not make sense for Intra or inter rater. Should it be summarized? You method may outline was is acceptable to extract and leave out what is not.

The COSMIN framework suggests that Correlation (Pearson or Spearman ) alone or t-test alone is not adequate to study reliability but both together are ok. Can you report how many used correlation alone, t-test alone and the combination.

Reliability indices are part of it but it would be important to extract estimates of measurement errors such as SEM, MAD, Bland and Altman bias and Limits of agreement. Specify how often those were available and extract when they were.

You have alternated between Intra investigator and intra researcher. I suggest you consistently use intra-investigator throughout.

L206 You had not announced what you deem adequate for CV%, Please do.

Section 3.4.1 (both Line 197 to 218) It is inappropriate to focus on reliability by studies as a whole. You should report it for specific measurements. Many measurements can be derived from ST and the reliability should only be aggregated for similar measurements. Not surface topography as a whole.

Correct section header on L208

L211 Specify here too when SCC is used with a test of difference in means or alone. (alone is not an adequate method to test reliability.

Table 4. The specific estimates for each measurement should be extracted. Standard for interpreting reliability require Mean and SD of the samples tested, the ICC type and values and error estimates if available. Documenting how often all the needed info was present would be important.

Correct typo to validity in table header.

Validity data extracted should include which pairs of measurements were contrasted and the reference standard method used to obtain reference measurements. A measure of associations or other validity statistics should be extracted on a per measurement basis. It does not make sense to summarize at the study level because some measurements may be have met standards and others been dismissed after these investigations. The relevant information is at the measurement level. With the latter level of details it will become possible to formulate recommendations on which measurements have the best properties and which should be abandoned and maybe identify which system produce common measurements the best.

Table 4 define all abbreviations in legend. EG. PD?

You may need to prepare an appendix with all the ST measurements definitions including what they are and how they are scored.

L225 Specify that 29 and 33 appear to have used both Correlation and t-test.

L230. This general rule may work well when comparing two measurements of the same thing using different methods. However it would be relevant to have different expectations when correlating one measurements on ST with a different measurement on x-rays which are not conceptually as similar. I disagree with a general approach here.

L231 to 236 This level of summary by studies is simply too general. We need a summary by specific measurement. Different studies have used different measurements and cannot be compared as you did.

I have summarily peeked at the discussion given it will likely change significantly by working at the level of measurements in the future.

Still It would be relevant to add an element. You chose to validate again X-ray and internal measurement which surface capture external posture changes. There may have been other approaches to consider to validate and questionnaire on perceived appearance may also reflect another aspect of validity. You need to justify why you chose x-ray and recognised that if ST is used not to replace radiograph but as an objective posture improvement capture technique then a different validation approach may be warranted.

I like that in the discussion you try to summarize by planes, landmark-based or 3D but this should be how the results are organized such that the discussion flows from the results.

In-text citations have the same problem as the table with using first names of authors rather than last names.

The discussion should clearly address that if ST measurements will be used to document changes over time in posture there should be evidence on the sensitivity to change and responsiveness of different parameters. Justify why you did not review this evidence in your systematic review and outline its importance.

The discussion should also aim to identify research gaps. Could you list measurements which have been insufficiently studied in terms of reliability validity? Are there system with insufficient evidence too? Revisit the important methodological quality issues and formulate recommendations to improve future research.

Author Response

(The authors gave the same response as above.)

Reviewer 3 Report

   The paper was well written and easy to follow.

   The tables are appropriate.

   The eligibility criteria, including inclusion and exclusion criteria, are well mentioned and appropriate.

   Some typos should be corrected (for example, the legend under the table 2.).

Author Response

(The authors gave the same response as above.)

Reviewer 4 Report

The authors screened 17 studies in this manuscript and analysed their reliability and validity. The search strategy, screening procedures, and eligibility criteria were clear. Two reviewers performed the screening procedures to improve the validity of this systematic review, and the third reviewer participated if there was controversy. However, here are some points to consider:

·         Since the authors mentioned that 12 of 17 studies were rated high quality, it is better to give more discussion regarding the 12 high-quality ones.

·         In the acquisition protocol part (Table 3), authors used multiple terminologies to represent surface topography (i.e., ‘ST acquisition’, ‘topographic measures’, ‘surface scans’). Although this makes the vocabulary of the article more diverse, it may confuse the readers.

·         In the discussion part, the authors also proposed some potential reasons for these studies that showed poor results. It is better to provide some assumptions for these studies showing satisfactory results. It helps others avoid similar experimental procedures.

·         Two opinions that the authors proposed in the discussion part were controversial. For instance, the authors thought the reason for poor validity is that “… the ST parameters and radiological parameters did not match…. There is poor correlation between maximal rotation and maximal cobb angle” (274-277). However, in the following paragraph, they proposed:” …, they can somewhat reflect the Cobb angle’s value.” (333-335). It is necessary to give a more straightforward conclusion, in which case the ST system is not matched to the Cobb angle, and in which case the ST system can reflect the Cobb angle.

Specific Comments:

·         Line 7: It would be better to introduce surface topography as “one of assessment methods” to orient the abstract readers on non-bias approach of authors.

·         Line 22: The conclusion of abstract highlights the potential of ST technique in treatment and management of scoliosis while the topic is mainly related to the assessment.

·         Line 53-54: The authors have sub-categorized methods to differentiate the 3D scanning from the surface topography. It is inconsistent with the 17 included papers in this study in which some have used the 3D scanners as the surface topography system.

·         Line 54-56: The justification of popularity of this method over other methods is not convincing as the 3D spine image is just based on the surface topography and not directly reflects the spine feature itself.

·         Line 98-99: Were the patients with spinal orthoses excluded based on the captioned exclusion criterium?

·         Line 158-159: The definition of adolescents may not include the subjects aged 20-25.

·         Line 185-207: Using inter/intra investigator and inter/intra researcher are inconsistent and make readers a bit confusing.

·         Line 208: Typo it seems to be 3.4.2. INTER-

·         Line 230: Typo Table 6

·         Line 231-232: The sentence should be improved.

·         Line 258-262: Knowing the fact that these are the parameters which might affect the accuracy of ST system, how do the authors support the validity of the ST as satisfactory?

·         Line 280:  The abbreviation “PC” is not introduced earlier in the text. What does it stand for?

·         Line 324: Inconsistency in reference citation. It should be probably Maria et al. Similar observation was found in other parts of the manuscript and the authors are advised to amend them.

·         Line 368-370: Again, it seems to be the guidelines to the authors rather than manuscript content and it should be removed.

Author Response

(The authors gave the same response as above.)

Round 2

Reviewer 2 Report

jcm-1951687 my re-review.

Abstract

I suggested some typo corrections in the abstract.

I suggested editing the sentence about Formetric to refer to its measurements.

In the abstract conclusion: Good common measurements is too vague.

Also the Last sentence may not be supported by evidence and should be presented as hypothetical. You did not collect data showing that the same clinical conclusion can be achieved by using ST rather than radiographs.

I have suggested typo and some minor edits in the intro.

L111. Please add a sentence to define what you mean by your Asymmetric measurement category. I recommend stating asymmetry parameters rather than asymmetric parameters throughout.

L173. Please report the agreement between reviewer 1 and 2 before inviting Reviewer 3 at the full-text stage. (report percent agreement and Kappa on the decisions to include or exclude.) or mention not doing so as a limitation.  Similarly could report agreement on extraction or on quality appraisal.

L191. Report what the criteria were to conclude that the heterogeneity was too important to conduct a meta-analysis. EG, less than 3 studies on same measurements or in same subject group or ….

The title for table 2 should be presented on the same page as table 2.

L206 207 Please add a sentence defining what you mean by asymmetry measurements.

L223 Your reliability standards and validity standards are fairly low. While you did cite references these standards would be such that a moderately reliable and valid statement can be assigned when less than half the variability is explained by the ST measurements. This is hardly adequate to suggest the ST measurements could replace x-rays. Cosmin does present guidance on standards. I had recommended you refer to those. Here you should either update those standard or add a careful paragraph outlining this limitation.

Even if you choose to only add a limitation please also expand in the discussion that you now are aware of COSMIN guidelines on conducting systematic reviews of measurements and that such documents could have provided guidance on designing the search and conducing a more thorough quality appraisal.  

L247 to 250 The rules on what is an appropriate interval should have been reported in the methods where you introduce the quality appraisal tool.  (and in legend of table 4?)

Table 4 should fit on the same page. Why have some quality appraisal scores been changed?

L369. Could report that other systems were reported in only 1 study each.

L388. Specify curve type rather than just type of scoliosis to avoid confusion with different diagnoses in this paragraph.

Formatting in table 5 will need adjustment as some line content stretched or cut. In many of the position descriptions in table 5 it would be best to state stood rather than stand I believe.

Align author name line 13 in table 5.

L615. I would recommend defining all abbreviations in the legend of the table. Forcing readers to download an appendix to understand the measurements tested is not practical.

L636 Why you do you refer to inter-investigator reliability in this section 3.4.1 under the Intra-investigator reliability.

L637 I recommended an edit to put emphasis on reliability of measurements. measurements

L641 and 642 again added reference to measurements to avoid suggesting that a study presented a certain level of reliability.

L647 to 649 Move this definition in intro for asymmetry parameters.

L659 Keep only the last mention of as the evaluation index.

L664. Again insert Their measurements before had good inter-investigator reliabilties. A reliability is a property of the measurements not the system.

L821. Insert “measurement derived from” before “automatic identification…

L825 Abbreviations used in the text should be defined on first use.

K830 and 831 Rather than referring to with one study showing. Please refer to the specific measurements  and cite the studies in this sentence. Again avoid suggesting that the reliability is a property of the system or the study, it is a property of a specific measurement.

Table 6. Should be left aligned rather than full justified for readability.

You should split table 6 into two tables 1 for reliability results and one for validity results. 

The reporting the list of indicators is a space efficient way to report but very hard to read. I recommend you have a Intra results column separate from the inter-investigator results.

SEM, SDC should be reported with measurement units where relevant. (degrees, mm,….) You have made odd use of ; and ,  and spaces in the reporting.

The abbreviation list is a good idea. I would like to have it part of the main text as a table rather than as an appendix.

The list of abbreviation should have a first colum listing only the abbreviations. The second the developed abbreviation and then the third column could report the description.

The list should be organized in alphabetical order of the abbreviations.

SA should specify this is an angle in the frontal plane.

TK and LL descriptions should specify this is an angle in the sagittal plane.

I am unclear from your description how VO amp differs from VO max.

RPC includes multiple values it is not clear from the description how you get to one parameter from all these steps. Is it the sum of the most important by areas?

C7 Plumb line arrow: I believe you need to specify that this is the horizontal Anterior-posterior distance.

For all the arrows it would be best to refer to horizontal distances for clarity.

Inflection point. I believe the explanation should refer to the vertical position of the inflection point.

Kyphotic and lordotic angles should specify these are in the sagittal plane.

Many parameters listed in this appendix do not have an abbreviations specified. Should they?

Only one maximum is described but the name of the parameters suggest there could be one per region. Please clarify: Thoracic/ Thoracolumbar gibbosity/ Paraspinal lumbar prominence

Thoracic/Thoracolumbar /Lumbar height : the description is unclear to me. Height of what aspect of the deformity. I am unclear if it is just referring to where it is located along the vertical C7-cleft line or also captures some aspect of the magnitude of the deformity there.

Quantec angle. Refer to this angle being located in the frontal plane I believe.

PD The explanation need to define GPA, what is a semi-landmark? Can we place these distance in a plane in particular?

For each parameter in this appendix specify if you classified it as frontal, axial, sagittal or asymmetry.

Asymmetry index needs to be explained here the reader should be able to have an idea of what this is without going to the original paper.

The appendix table should include citations where you pulled your description from.

 Similarly all the list here should be defined in the current table: Maximum anterior shift, Maximum anterior shift, Anterior/posterior range, Maximum right shift, Maximum left shift, Right/left range, Maximum CCW rotation, Maximum CW rotation, Rotation range, Smallest residual, Largest residual, Residual range, Minimum right/left asymmetry, Maximum right/left asymmetry, Right/left asymmetry range

The definition presented as: Kyphosis and lordosis are measured using the surface tangents of the inflection points of the convex and concave curves.  Is equivalent to what has been reported for TK and LL, Coombine into one.

The definition is insufficient for Coronal vertebral Axis, Sagittal vertebral axis, Pelvic obliquity. Please list separately.

For each measurement try to report the measurement units.

For trunk inclination specify if this measured as an angle or as a distance.

For back surface rotation and axial trunk rotation: Define the orientation of the x-axis as vertical horizontal or Anterior-posterior.

Deviation of the curve. Specify if it is measured as a distance in medio lateral , AP or in any direction in the transverse plane.

For DHOPI the distance I should be defined.

In the PC definition It seems to me than anterior is used when previous could be used instead.

Reliability reporting standards would suggest interpreting the reliability data in table 6 along with the means and SD reported on each measurement. This could be added to the reliability table.

Limiting the reporting of ICC to 2 decimals places and the reporting of ST to a reasonable but adaptable number of Decimals would help simplify the table.

Table 6 line 6. You report they used linear regression yet your only report r?

Table 6 line 7. You have two entries for maximum anterior shift with different results and no max posterior schift. Please check.

CCW and CW not defined in appendix.

In all the cell of table 6 please organize results by plane. For some studies you did specify but not for all. Once limited to just reliability maybe grouping results from a specific plane in different columns would be valuable for the readers.

Table 6 for Manca et al. ICT-ILL and ITL-ILS not defined in the appendix.

Table 6 for example Pazos. It would be more intuitive to order the parameter the same as you list them in column 2. Here superscript 3 is different than ICC, TEM and SDD.

Table 6 16  You list inter followed by Intra but in some other placed you did intra first. I recommend always  reporting Intra first in the table when both are available. For this entry under validity add an explanation for why they are two r values presented for each analysis.

Similarly for Yildrium results why are there 2 correlations reported for each validity analysis.

You always specify what stats was reported in the first column under validity. Then in the results you sometimes state r= or just the value. Pick one and be consistent throughout.

Line 1128 You wrote: Except in 1128 one study[38], two ST measurements (lumbar scoliosis curve and sagittal vertebral axis) 1129 showing moderate correlation occurred.

Instead I suggest: There was one study exception [38] where two ST measurements (lumbar scoliosis curve and sagittal vertebral axis) showing moderate correlation occurred.

L1138 I added between corresponding measuremetns at the end of the formetric sentence.

L1150 Add measurements: Formetric measurements obtained.

L1154. Replace system by measurements.

Why are you singling out Formetric? It does seems you may have a conflict of interest pushing this system over others.

L1163. And L1374 Replace system by measurements.

L1375 replace regions by measurements.  In the parenthesis name the specific problematic measurements)

L1415 Replace paste by placement?

L1440  to 1470 This paragraph is presenting contradicting idea. On one hand you suggest Cobb angle correlation are not important but then you switch back to saying they are of interest to resaerchers.

This paragraph would be a good occasion to discuss if your validity standards were high enough to allow intercheanging the X-ray and ST or not. One would want high correlations or even better high intermethod ICC to determine if the different measusrements can be used interchangeably in testing patients. Please discuss how much evidence was strong enough where interchanging may be possible. Or if not enough evidence outline what is missing.

L2435. I suggest rephrasing as follows However, for measurements from some other common systems, such as InSpeck and Quantec the evidence was not as conclusive due to the limited number of included studies.

In the discussion. I would like a paragraph making recommendations on how to improve the common methodological weaknesses detected by your quality appraisal.

I would also want a paragraph possibly listing a number of parameters which have been only tested for some but not all 3 properties reviewed (intra- or inter reliability, and validity). It would be useful to present a research agenda for the missing information.

I am not sure what you mean by good and by common in the following sentence. Please rephrase: The Formetirc system can provide good common measurements.

You did not review study comparing if clinical decision would be similar when presented with only ST or x-ray to arrive at the conclusion here: This review suggests that the ST technique has great potential in assessing scoliosis, especially in reducing radiation exposure and performing cosmetic assessments.

Please remove or rephrase as hypothetical or suggest what kind of research is now needed given the promising reliability and validity results you found.

I have uploaded a file with typo and edit correction. Go to the very end for my edits on the appendix pages. 

Author Response

Date: Nov 21, 2022

Dear reviewer,

Thank you for your second round of comments concerning our manuscript entitled"Reliability and validity of scoliosis assessment using surface topography techniques: A systematic review"(Manuscript ID: jcm-1951687). Those comments are all valuable and very helpful for revising and improving our paper and the essential guidance for our further research. We have studied the comments carefully and made corrections, which we hope meet with approval. Revisions to the manuscript were marked up using the “Track Changes” function, and the detailed responses to your comments are uploaded as a pdf attachment.

Special thanks to you for your good comments.

Sincerely,

Ye Liu

[email protected]

School of Sport Science

Beijing Sport University, Beijing, China
